# Changes in Soil Aggregate Carbon Components and Responses to Plant Input during Vegetation Restoration in the Loess Plateau, China

**DOI:** 10.3390/plants13172455

**Published:** 2024-09-02

**Authors:** Yaoyue Liang, Jingbo Fang, Wenjing Jia, Shijie Wang, Hanyu Liu, Weichao Liu, Qi Zhang, Gaihe Yang, Xinhui Han, Guangxin Ren

**Affiliations:** 1College of Agronomy, Northwest A&F University, Xianyang 712100, China; liangyaoyue@nwafu.edu.cn (Y.L.); fangjingbo@nwafu.edu.cn (J.F.); 2022050001@nwafu.edu.cn (W.J.); 550530906@nwafu.edu.cn (S.W.); hy-liu@nwafu.edu.cn (H.L.); liuweichao1014@163.com (W.L.); 178542993@163.com (Q.Z.); ygh@nwsuaf.edu.cn (G.Y.); 2The Research Center of Recycle Agricultural Engineering and Technology of Shaanxi Province, Xianyang 712100, China

**Keywords:** soil aggregate, organic carbon components, organic functional groups, vegetation restoration, Loess Plateau

## Abstract

Vegetation restoration is an effective measure to cope with global climate change and promote soil carbon sequestration. However, during vegetation restoration, the turnover and properties of carbon within various aggregates change. The effects of plant source carbon input on surface soil and subsurface soil may be different. Thus, the characteristics of carbon components in aggregates are affected. Therefore, the research object of this study is the *Robinia pseudoacacia* forest located in 16–47a of the Loess Plateau, and compared with farmland. The change characteristics of organic carbon functional groups in 0–20 cm, 20–40 cm, and 40–60 cm soil layers were analyzed by Fourier near infrared spectroscopy, and the relationship between the chemical structure of organic carbon and the content of organic carbon components in soil aggregates was clarified, and the mechanism affecting the distribution of organic carbon components in soil aggregates was revealed in the process of vegetation restoration. The results show the following: (1) The stability of surface aggregates is sensitive, while that of deep aggregates is weak. Vegetation restoration increased the surface soil organic carbon content by 1.97~3.78 g·kg^−1^. (2) After vegetation restoration, the relative contents of polysaccharide functional groups in >0.25 mm aggregates were significantly reduced, while the relative contents of aromatic and aliphatic functional groups of organic carbon were significantly increased. The opposite is true for aggregates smaller than 0.25 mm. (3) With the increase in soil depth, the effect of litter on organic carbon gradually decreased, while the effect of root input on the accumulation of inert carbon in deep soil was more lasting.

## 1. Introduction

Soil aggregates are the basic units and components of soil organic carbon, closely related to soil organic carbon content. Therefore, soil aggregates are of great value for soil carbon sequestration [1,2]. The organic carbon structure of soil is a reflection of the stability and quality of organic carbon, and one of the primary factors influencing soil carbon sequestration is thought to be its resistance to microbial degradation. A different soil organic carbon structure is the root cause of soil organic carbon stability differences, which is the main influencing factor of carbon turnover in soil by indicating the substrate characteristics of carbon [3]. The types and contents of soil organic carbon functional groups constitute the chemical structure of soil organic carbon and thus characterize the activity inertness of soil organic carbon [3,4]. Restoring vegetation is essential to preserving soil stability and improving soil carbon storage [5,6]. Vegetation restoration mainly involves the input of fixed carbon by litter and root exudates into soil organic carbon pools. However, current studies on the effects of plant source carbon input on soil organic carbon mainly focus on the perspective of whole soil [7,8], and the differences in the turnover and characteristics of soil aggregate organic carbon and its driving factors are unclear. Therefore, the aim of this study was to investigate the effect of plant source carbon on the change and turnover of organic carbon components in aggregates. It provides a scientific basis for the fixation mechanism of organic carbon in soil aggregates in fragile habitats of the Loess Plateau.

Large aggregates (>2 mm), medium aggregates (0.25–2 mm), microaggregates (0.053–0.25 mm), and clay particles (<0.053 mm) are the general categories into which aggregates can be separated based on their particle size. Different aggregates with varying particle sizes have varied organic carbon contents [9]. Active sites rich in medium aggregates are more able to adsorb fixed organic carbon, and new organic carbon is enriched in medium aggregates due to better physical and microbial protection. Large-particle-size aggregates contain relatively high organic carbon content but unstable carbon components, while small-particle-size aggregates have lower organic carbon content but more stubborn C substrates [10,11,12]. Studying the organic carbon content of soil aggregates of different particle sizes can effectively reveal the coupling relationship between the stability of soil aggregates and soil organic carbon, so it is of great significance to understand the change characteristics and mechanism of soil organic carbon content of aggregates of different particle sizes and its components.

Chemical structure is an important internal factor affecting the quality and function of organic carbon. The structure of organic carbon is complex, mainly composed of carboxyl, hydroxyl, ether, and other functional groups. The relative abundance of different functional groups of soil organic carbon has a profound influence on its chemical reactivity and stability [13,14]. Fourier infrared spectroscopy (FTIR) can measure the chemical functional groups of soil organic carbon to reflect its chemical composition, structural changes, and other information [15]. The input of foreign carbon can be used as a cementing agent to change the composition of soil aggregates, thus changing the distribution of soil organic carbon [12,16,17]. While the free organic matter that is not enclosed by the aggregates is more easily broken down and mineralized, the soil aggregates use the feedback mechanism to isolate the recently imported organic matter in space and create physical protection [18]. Moreover, exogenous carbon input, such as litter, will cause a strong excitation effect [19]. Due to the complex changes in the environment caused by vegetation restoration, the contribution of litter and plants to the composition of carbon structure in different aggregates is still unclear.

Based on this, the research object for this study was retired *Robinia pseudoacacia* forest soil of various years and widely planted corn farmland in the Loess Plateau. This study examined changes in aggregate stability as well as organic carbon content, components, and chemical structure during the vegetation restoration process from the viewpoint of soil profile aggregates. The response of soil aggregate stability, organic carbon composition, and organic carbon structure in different soil layers to different years of arable land was determined. The influence mechanism of vegetation–soil characteristics on organic carbon components of aggregates in the reverted *Robinia pseudoacacia* forest was elucidated. This took place in order to provide a theoretical reference for the change mechanism of aggregate organic carbon in the fragile habitat area of the Loess Plateau, and provide a scientific basis for the sustainable management of the agroforestry system in the Loess Plateau.

## 2. Materials and Methods

### 2.1. Overview of the Study Area

In this study, the Wuli Bay basin (109.19–109.22° E, 36.51–36.53° N) in Ansai, northern Shaanxi Province, was selected, with an altitude of about 1060–1370 m. This area experiences a warm temperate semi-arid monsoon climate, with an average yearly temperature of 8.8 °C and an average yearly precipitation of roughly 530 mm, which is primarily concentrated in July through September. The annual frost-free period is approximately 165 days, and the annual sunshine hours are roughly 2350–2570 h. The soil in this basin is loose and barren, and the soil type is yellow spongy soil with low fertility formed by the development of loess parent material, poor water retention and erosion resistance, and low carbon storage. After decades of a vigorous implementation of ecological construction projects such as vegetation restoration (grass) and soil and water conservation, the ecological environment of the basin has improved significantly, and the vegetation coverage rate has increased significantly, from 17.70% before the return of farmland to 45.50%. The forest is mainly composed of the tree *Robinia pseudoacacia* and the shrub *Caragana intermedia*.

### 2.2. Sample Site Setting

By consulting the relevant literature and combining data on vegetation restoration provided by the local government, in July 2021, farmland (corn) and artificial *Robinia pseudoacacia* forest soil with restoration years of 16 years (since 2005), 22 years (since 1999), 32 years (since 1989), and 47 years (since 1974) were selected as the research objects in Wuliwan Watershed. The linear distance between various fields was less than 2 km. Only the restoration years of the selected plots were different, and other characteristics such as site conditions and soil texture were similar. During the period of vigorous plant growth (July–September), litter, roots of understory herbs, and soil samples were collected. Three plots were selected in each year for the *Robinia pseudoacacia* plantation, and a standard sampling area was set up in each plot.

### 2.3. Screening of Soil Aggregates

Dry sieving of soil aggregates: The aggregate vibrating screen was used for the dry screening of soil aggregates. The broken soil blocks were transferred to the upper layer of the sieve (2 mm, 0.25 mm, 0.053 mm), the frequency was set to 200 times per minute, the oscillation time was 1 min, the cover was closed, and the instrument was started, and the aggregate soil at all levels of the sieve was removed after the dry screen was finished and weighed, respectively. This is utilized to assess an aggregate’s mechanical stability. Wet sieving of soil aggregates: Wet sieving of soil aggregates is performed using an aggregate analyzer. The sample was placed on the top layer of the aggregate analyzer composed of the pore size of 2 mm, 0.25 mm, and 0.053 mm, respectively, with the frequency set to 30 times/min and the oscillation time to 30 min. After the screening, the soil aggregates in each layer of the screen were washed into the tin foil box to obtain aggregates with a particle size of >2 mm, 0.25–2 mm, 0.053–0.25 mm, and <0.053 mm, respectively. After drying, they were weighed, and the percentage of aggregates of each particle size and the water stability of aggregates were calculated for the determination of the organic carbon content of subsequent aggregates [20].

### 2.4. Data Analysis

The changes in soil physicochemical properties, carbon pool components, and functional group characteristics of aggregates during *Robinia pseudoacacia* restoration were treated by one-way ANOVA, and the least significant difference (LSD) method was used for multiple comparison (a = 0.05). The relationship between carbon pool components and carbon functional groups of aggregates was used by a Pearson correlation analysis. Lastly, the reactions of aggregate organic carbon and plant source carbon input in various soil layers were investigated using variance decomposition and a redundancy analysis. The data were sorted out by Excel 2021 and analyzed by SPSS 27.0. Then, R language (4.2.3) and Origin 2021 were used to complete the mapping.

## 3. Results and Analysis

### 3.1. Change Characteristics of Soil Organic Carbon Components in Different Years of Tillage

The MWD and GMD of topsoil farmland were significantly higher than forest land, and there was no significant difference among all years of *Robinia pseudoacacia* forest land. There was no significant difference in MWD among different plots in deep soil, but GMD increased with the increase in vegetation restoration years, and the GMD content of *Robinia pseudoacacia* was significantly higher than that of farmland in 47 years (Appendix A). The changes in soil organic carbon components and the change characteristics of soil aggregate particulate organic carbon and soil aggregate mineral-bound organic carbon in different years of arable land are shown in Figure 1. As soil depth increased, aggregate particulate organic carbon and mineral-bound organic carbon as well as the soil’s organic carbon content often declined. In addition, particulate organic carbon and mineral-bound organic carbon showed similar trends for each particle size in each soil layer. The trend for the organic carbon in the 0–20 cm soil was initially declining and then increasing, with a change range of 1.97–3.78 g·kg^−1^. Furthermore, the content of medium aggregates increased as the number of years of return increased, while the content of microaggregates remained relatively constant and declined with the number of years of return for large aggregates. The particulate organic carbon and mineral-bound organic carbon of 20–40 cm soil organic carbon and aggregate of each grain level increased first and then decreased, and reached the highest level in 22 years after the return of tillage. The particulate organic carbon and mineral-bound organic carbon of organic carbon as well as the aggregate of each grain level in the 40–60 cm soil layer likewise showed a tendency of first growing and then dropping. Similarly, the particulate organic carbon and mineral-bound organic carbon of each particle size aggregate increased at first and then decreased with the increase in the age of return, and the contents were the highest at 32 years of return.

### 3.2. Modifications in the Properties of the Functional Groups of Organic Carbon in Soil Aggregates with Varying Tillage Years

Generally, in >0.25 mm aggregates, plant restoration greatly raised the relative amounts of organic carbon aromatic and aliphatic functional groups and significantly decreased the relative contents of polysaccharide functional groups, and reversed in <0.25 mm aggregates (Figure 2). Thus, the stability of >0.25 mm aggregate organic carbon is improved, and the stability of <0.25 mm aggregate organic carbon is decreased (Appendix A).

In the 0–20 cm soil layer and farmland, the relative content of the aromatic C=O functional group gradually increased with the decrease in grain size. In the *Robinia pseudoacacia* forest, the relative content of polysaccharide C-O functional groups in RP47 was the highest in clay particles, and the relative content of polysaccharide C-O functional groups in other years of the *Robinia pseudoacacia* forest was the lowest in microaggregates. In the 20–40 cm soil layer and farmland, the relative content of polysaccharide C-O functional groups was the highest in aggregates with a diameter >0.25, and the relative content of methylene CH_2_ deformation and vibration functional groups was higher in medium aggregates than other grains. In the *Robinia pseudoacacia* forest, the relative content of the carbohydrate C-O functional group and methylene CH_2_ deformed vibration functional group was the highest in the medium aggregate. In the 40–60 cm soil layer, the relative content of C-O functional groups increased with the decrease in the particle size in RP16, and the relative content of C-H deformed vibration functional groups in RP47 was the highest in <0.25 mm particle size aggregates in the *Robinia pseudoacacia* forest.

The correlation analysis between organic carbon components of soil aggregates and organic carbon structure of aggregates in different years of the afforestation restoration sequence is shown in Figure 3. The organic carbon in aggregates was significantly correlated with the C-H deformation vibration of polysaccharide C-O, fat C, aromatic C=O, and OH stretching vibration functional groups of phenolic compounds (*p* < 0.05). In 0–20 cm and 40–60 cm soil layers, SOC content in aggregates with a particle size > 0.25 mm was significantly positively correlated with mineral-bound organic carbon content. There was a significant positive correlation between SOC content and microbial biomass carbon content in clay particles. The content of microbial biomass carbon was positively correlated with the content of dissolved organic carbon in partial-particle-size aggregates and negatively correlated with the content of dissolved organic carbon in partial-particle-size aggregates.

### 3.3. Response of Organic Carbon Components to Plant Input in the Aggregates of the Reverted Robinia pseudoacacia Forest

To study the effect of plant input on soil organic carbon components, variance decomposition was further obtained, and soil organic carbon and particulate organic carbon contents of topsoil were mainly affected by litter. While the influence of litters on soil organic carbon gradually weakened and the residual was gradually reduced, the impact of roots on soil organic carbon grew progressively deeper as soil depth increased. In the deep soil layer, it was mainly the joint action of roots and litter that affected the content of soil organic carbon, and the influence was gradually deepened. Particulate organic carbon in the surface layer is mainly affected by litter, and then the effect is rapidly reduced and gradually transformed into a deeper effect of roots. Both roots and litter play an important role in the formation and stability of mineral-bound organic carbon.

## 4. Discussion

### 4.1. Changing Characteristics of Soil Aggregate Components with Different Years of Tillage

Soil organic carbon is mainly derived from vegetation’s litter, root residues, and secretions [21]. Inputs from surface litter, roots, root exudates, and root debris lead to increasing organic carbon content in the surface layer [22]. Similar to the results of this study, the results showed that vegetation restoration significantly increased the organic carbon content of surface soil aggregates, with a range of 1.97–3.78 g per kilogram. As the years of vegetation restoration increase, plant population structure becomes more complex, plant source carbon input gradually increases, and more plant residues return to the soil and are decomposed by microorganisms, resulting in an increase in soil surface organic carbon content. The total organic carbon content in large and medium aggregates in the 20–40 cm soil layer increased and subsequently dropped, whereas the concentration in small aggregates and clay particles first decreased and then climbed. The content of >0.25 mm aggregates was the highest, while the content of <0.25 mm aggregates was the lowest. It may be that the small aggregates and clay particles in the early stage of vegetation restoration formed large aggregates due to the interconnection of roots, mycelium, and calcium carbonate, and then the decomposition of large aggregates in the late stage of vegetation restoration increased the content of microaggregates and clay particles [23]. In the 40–60 cm soil layer, the organic carbon content of <0.25 mm aggregates first increased and then decreased, and the contents of microaggregates and clay particles continuously changed, which may be the result of the comprehensive influence of multiple factors such as environmental and climatic conditions, land use patterns, and vegetation management measures.

The aggregate particulate organic carbon of the various years in this study demonstrated a trend of first rising and then falling with the decrease in grain size. Macroaggregates contain higher concentrations of organic carbon than microaggregates, which may be due to the dominance of fungi in macroaggregates, resulting in high concentrations of organic carbon and microbial biomass carbon [24]. The dynamic change in soil particulate organic carbon content depends on the balance between the formation of particulate organic carbon by the input of plant residues and the loss of particulate organic carbon through microbial decomposition. In the topsoil, particulate organic carbon and mineral-bound organic carbon were significantly higher than those in the soil after returning to farmland. It may be that farmland promotes carbon accumulation due to increased nitrogen enrichment through fertilization, field management, and other measures. Excessive nitrogen application will lead to soil acidification, inhibit the decomposition of stable carbon, promote the formation of soil humus and stable carbon, and also lead to more carbon entering the soil from litter and root exudates, resulting in soil carbon accumulation and increased particulate organic carbon and mineral-bound organic carbon reserves [25]. The results showed that mineral-bound organic carbon was mainly stored in clay particles (<0.053 mm), and clay particles could not only physically isolate the old carbon available in the interior from microorganisms, but also the microbial residues in the interior could strengthen the binding of soil minerals to form mineral-bound organic carbon and remain in the soil relatively stably [12,26]. The slow carbon pool of mineral-bound organic carbon is more sensitive in the process of vegetation restoration, which may be caused by many factors such as terrain, soil conditions, and microorganisms.

### 4.2. Variations in the Properties of the Functional Groups of Organic Carbon in Soil Aggregates with Varying Tillage Years

The variations in the chemical composition and relative concentration of soil organic carbon in soil aggregates with different particle sizes had an effect on the turnover and fixing of soil organic carbon. On the whole, the aliphatic carbon content was higher and the activity was stronger in the >0.25 mm aggregate. In the <0.25 mm aggregate, the aromatic carbon with strong decomposition resistance will selectively accumulate, the activity is weak, and the stability is strong. In the three soil layers, the relative content of aromatic C=O functional groups in the aggregates >0.25 mm after 47 years of cultivation was higher than that in farmland, while the relative content of polysaccharide C-O functional groups was lower than that in farmland, indicating that the long-term cultivation of *Robinia pseudoacacia* increased the relative content of aromatic organic carbon in the soils of large aggregates. They can be used by soil microorganisms to attach extracellular polymers to mineral surfaces but reduce the accumulation of carbohydrates in soil aggregates with large particle sizes. The litter and its secretions accumulate gradually in the soil, and the decomposed product lignin contains a lot of aromatic compounds, which leads to the accumulation of aromatic functional groups in the large-particle-size aggregates in the soil. Lignin is generally considered an important source of a slow-acting carbon pool due to its difficult degradation [27,28]. In this study, afforestation increased the relative content of the C-H deformation vibration of fat C in aggregates. The reason may be that vegetation restoration leads to an increase in plant source carbon input in soil active components [3].

The stability of large aggregates rose with vegetation recovery in the three soil layers, while the stability of clay particles decreased as vegetation recovery time increased (Appendix A). This is mainly due to the fact that with the recovery of vegetation, the formation and decomposition of organic carbon have been continuously continued by the cementation of clay particles with fine roots and mineral particles to form medium aggregates and then coalescence into large aggregates, so its stability always shows a declining state. At the same time, the organic carbon of large aggregates is mainly composed of light particles, particulate organic carbon [16], which have relatively low density and are not tightly bound to clay minerals, and are easy to mineralize. Due to the formation of clay particles and microaggregates, the stability of the medium aggregates is constantly increased, while the stability of the medium aggregates fluctuates greatly because they are intermediates in formation and agglomeration. The stability of aggregates with varying particle sizes in 40–60 cm soil was not significantly different from that of 0–40 cm soil. The reason was that the deep soil lacked fresh carbon sources and energy, resulting in low microbial activity, and limited the decomposition and formation ability of organic carbon, and the turnover of organic carbon tended to be stable [29].

In this study, the correlation analysis between the organic carbon components of soil aggregates and the organic carbon structure of soil aggregates showed that the organic carbon of aggregates was significantly correlated with the functional groups of polysaccharide C-O, fat C, aromatic C=O, and phenolic compound -OH stretching vibration (*p* < 0.05). The results showed that soil organic carbon content in aggregates of a >0.25 mm particle size was significantly positively correlated with mineral-bound organic carbon content in 0–20 cm and 40–60 cm soil layers, indicating that the organic carbon content in large aggregates was mainly affected by mineral-bound organic carbon, and the increase in stable organic carbon contributed to the increase in the soil organic carbon pool in aggregates of a >0.25 mm particle size. There was a significant positive correlation between soil organic carbon content and microbial carbon content in the clay particles, indicating that the organic carbon in the clay particles mainly came from microorganisms. The content of microbial biomass carbon was positively correlated with the content of dissolved organic carbon in partial-particle-size aggregates and negatively correlated with the content of dissolved organic carbon in partial-particle-size aggregates. This outcome might be explained by the fact that dissolved organic carbon is the primary source of nutrients for the growth activities of soil microorganisms, and that increasing its content will boost these organisms’ activity and raise soil respiration and microbial biomass carbon content. After the microorganisms consume a large amount of dissolved organic carbon for their own respiratory consumption, the dissolved organic carbon content may decrease.

### 4.3. Response of Organic Carbon Components to Plant Input in Aggregates of Reverted Robinia pseudoacacia Forest

Vegetation restoration diversifies plant community structure through the principles of niche differentiation and complementarization, and increases litter quantity and input types to cause changes in the soil carbon pool [30]. After vegetation restoration, the increased input of root exudes, such as organic acids, the main component of root exudes, can not only stimulate microbial activity to promote the turnover of soil organic carbon by inducing an excitation effect but also change the structure of soil organic carbon by affecting the complexation and dissolution of organic carbon. With the increase in the soil depth, the joint influence of litter and roots on organic carbon gradually increased (Figure 4). Litter contains a large amount of organic matter, and with the slow process of vegetation restoration, the organic carbon content of the soil surface increases rapidly, but the impact of litter on organic carbon decreases with the increase in soil depth. Plant root input has a more far-reaching impact with the increase in soil depth, mainly because roots have a longer-term impact on the accumulation of inert carbon in deep soil, and carbon formed by root activities is more easily protected by physical and chemical protection from microbial decomposition. The organic carbon components of aggregates in the 20–40 cm soil layer were significantly positively correlated with the lignin content in litter and roots (Appendix A), and the difficult decomposition of lignin ensured the effective carbon retention in soil. However, the primary factor affecting particulate organic carbon on the top layer is litter, which has a quick diminishing effect before gradually changing into a deeper root effect. The main reason is that the formation process of particulate organic carbon is the initial way for organic matter and nutrients in litter to enter the soil. It is mainly composed of relatively undecomposed light plant materials, sensitive to environmental changes with high turnover. The effect of the root system on particulate organic carbon is mainly related to root biomass and particulate organic carbon formation. The decomposition of particulate organic carbon involves a variety of soil processes and functions, in which the root system and its secretions play an integral role. Litter and root systems had different effects on mineral-bound organic carbon in each soil layer. From the perspective of mineral-bound organic carbon formation, mineral-bound organic carbon is thought to be formed mainly from microbial residues through chemical bonding with minerals [8,31]. A low-molecular-weight C substrate (LMW-C) is the material basis contributing to mineral-bound organic carbon. The turnover of underground plant residues such as rhizosphere deposit C and root residues is dominant in the formation of LMW-C [32]. However, a small portion of the above-ground LMW-C mainly comes from the leaching precipitation of the litter layer, humus layer, and interfacial soil layer. However, due to microbial metabolic consumption and leaching diffusion, the amount of LMW-C entering the soil decreases step by step. As a result, roots and litter are crucial to the development and stability of mineral-bound organic carbon.

### 4.4. Limitations

In this study, the artificial *Robinia pseudoacacia* forest of different years and the return of farmland in the Loess Plateau area were selected as the research object. The research object could not represent the functional group structural characteristics and the distribution law of organic carbon components in soil aggregates in other regions or other restoration modes. The applicability of functional group structural characteristics and the distribution law of organic carbon components in aggregates in other regions need to be further studied. In this study, only the correlations between aggregate organic carbon and aggregate functional group structure and vegetation–soil properties were analyzed. It has not studied its internal relationship with the mechanism, and it can be further studied. The functional group structure of organic carbon in aggregates can only be studied semi-quantitatively by Fourier infrared spectroscopy. A further analysis and comparison should be combined with other quantitative analysis techniques to provide the theoretical basis for the structural characteristics of organic carbon functional groups in aggregates.

## 5. Conclusions

In this study, the change mechanism of soil organic carbon characteristics was studied from the perspective of the soil layer and aggregate niche. In the Loess Plateau, vegetation restoration increased the surface soil organic carbon content and the active organic carbon components in the soil. Large and medium aggregates showed a drop in the relative proportion of polysaccharide functional groups and an increase in the content of aromatic functional groups. Variance decomposition further showed that with the increase in soil depth, the effect of litter on organic carbon gradually decreased, while plant root input had a more long-term effect on the accumulation of inert carbon in deep soil. It offers a solid scientific foundation for further investigation into the internal mechanism of soil carbon accumulation during plant restoration from the standpoint of aggregates with varying particle sizes. In addition, FTIR technology has made a new contribution to carbon sequestration in the Loess Plateau. The results of this study have reference value for revealing the response of carbon components of soil aggregates to plant input in the Loess Plateau, and have important significance for influencing the soil carbon cycle pattern of vegetation restoration.

## Figures and Tables

**Figure 1 plants-13-02455-f001:**
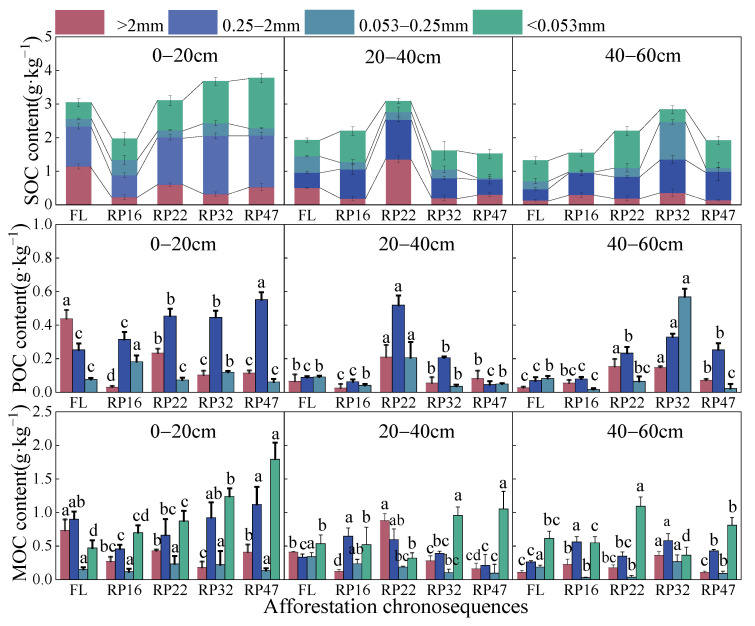
Content of organic carbon, particulate organic carbon, and mineral-bound organic carbon in aggregates of different grades in different years of *Robinia pseudoacacia* forest. Note: FL, RP16, RP22, RP32, and RP47 in figure represent farmland, *Robinia pseudoacacia* forest at 16 years, *Robinia pseudoacacia* forest at 22 years, *Robinia pseudoacacia* forest at 32 years, and *Robinia pseudoacacia* forest at 47 years, respectively. Lowercase letters a,b,c,d indicate significant differences between plots of different afforestation years (*p* < 0.05).

**Figure 2 plants-13-02455-f002:**
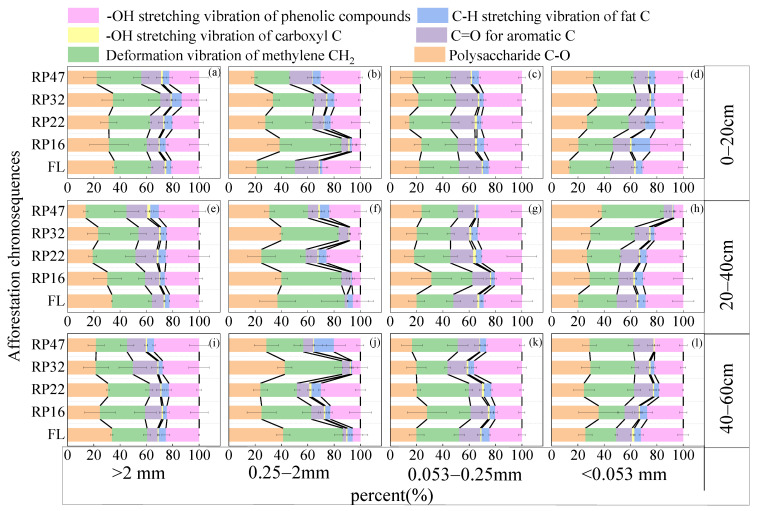
The relative content of the organic carbon functional groups in the aggregates of various grain grades in different years of the *Robinia pseudoacacia* forest. Note: FL, RP16, RP22, RP32, and RP47 in the figure represent the farmland, *Robinia pseudoacacia* forest at 16 years, *Robinia pseudoacacia* forest at 22 years, *Robinia pseudoacacia* forest at 32 years, and *Robinia pseudoacacia* forest at 47 years, respectively.

**Figure 3 plants-13-02455-f003:**
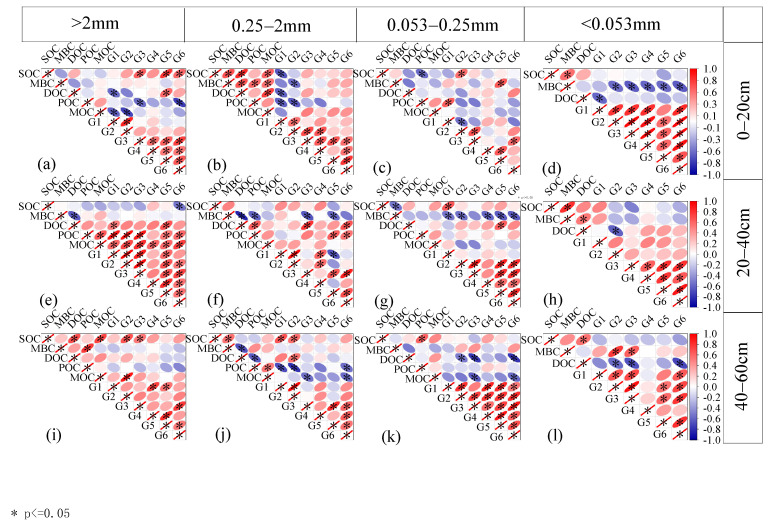
The correlation analysis of organic carbon components and organic carbonation structure of soil aggregates of *Robinia pseudoacacia* at different ages. Note: * represents a significant difference at the 0.05 level. G1 represents polysaccharide C-O, G2 represents the deformation vibration of methylene CH_2_, G3 represents the C=O of aromatic C, G4 represents the -OH stretching vibration of carboxyl C, G5 represents the C-H stretching vibration of fat C, and G6 represents the -OH stretching vibration of phenolic compounds.

**Figure 4 plants-13-02455-f004:**
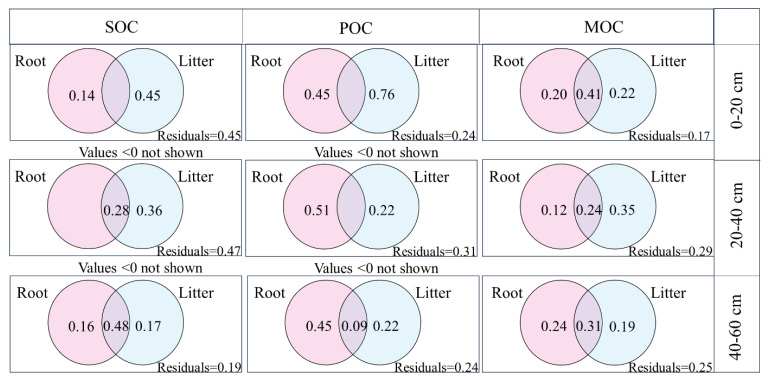
Variance decomposition of organic carbon component litter and root input in soil aggregates of *Robinia pseudoacacia* forest at different ages.

## Data Availability

Data are contained within the article and Appendix A.

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
