# Peer review of "Changes in Soil Aggregate Carbon Components and Responses to Plant Input during Vegetation Restoration in the Loess Plateau, China"

_plants, 2024, doi:10.3390/plants13172455_

Round 1
Reviewer 1 Report
Comments and Suggestions for Authors
The authirs provides a valuable analysis of how vegetation restoration affects soil organic carbon (SOC) components across soil layers and aggregate sizes. Focusing on the ecologically significant Loess Plateau, the study addresses a key gap in understanding plant input effects on SOC in soil aggregates. The authors' use of Fourier Transform Infrared (FTIR) spectroscopy to examine organic carbon functional groups is a notable strength, offering detailed insights into SOC stability and transformation. The findings on SOC responses across soil depths and aggregate sizes are particularly impactful. However, the MS has some weaknesses that should be addressed before publication. Firstly, the study's findings may not be generalizable to other regions with different climatic and soil conditions, and a discussion on the potential limitations of applying these results to other geographical areas or ecosystems would be beneficial. Secondly, while the paper establishes correlations between SOC components and functional groups, it lacks deeper mechanistic insights into the processes driving these changes. Additionally, the MS could improve by discussing the long-term implications of vegetation restoration on carbon sequestration and ecosystem stability, which would enhance the overall relevance and impact of the study. i would also suggest the authors can raad cite some good papers in the revised version if possible or applicable. DOI: 10.3390/rs15235571; DOI: 10.3390/rs13071279; DOI: 10.1016/j.agee.2022.108180; DOI: 10.1109/JSTARS.2023.3322640ï¼› furthermore, the ms is heavily focused on the scientific analysis of SOC components, but it could benefit from a more explicit discussion on the practical implications of the findings. For instance, how can these results inform land management practices, carbon sequestration strategies, or policy decisions related to vegetation restoration in the Loess Plateau or similar regions? in concluison, pl note that the authors can highlight the novel contributions of this study, particularly in terms of the methodological approach (e.g., use of FTIR) or the specific findings related to carbon sequestration mechanisms in the Loess Plateau.
here is some minor revison for your considering.
line 11 change into the turnover and properties of carbon within various aggregates change
line 12-13 consider simplifying for readabilityï¼›
line 35-36 change into "Soil aggregate" to "Soil aggregates are the basic units and components of soil organic carbon, closely related to soil organic carbon content.”
Line 52-54 simplify the sentence
Line 304 consider changing "big" to "large" for consistency
Some sentences could be clearer and more concise for better readability. Overall, I believe this is a good piece of work, and I would recommend minor revisions before it is accepted.
Author Response
|
1. Point-by-point response to Comments and Suggestions for Authors Comments 1: Firstly, the study's findings may not be generalizable to other regions with different climatic and soil conditions, and a discussion on the potential limitations of applying these results to other geographical areas or ecosystems would be beneficial. |
|
Response 1: Thank you for pointing this out. We agree with this comment. Therefore, I have added in line 384-390 of section 4.4 Limitations the potential limitations of the application of this study to other geographic regions or ecosystems. [In this study, artificial Robinia pseudoacacia forest of different years after the re-turn of farmland in the Loess Plateau area were selected as the research object. The re-search object could not represent the functional group structural characteristics and the distribution law of organic carbon components in soil aggregates in other regions or other restoration modes. The applicability of functional group structural characteristics and the distribution law of organic carbon components in aggregates in other regions needs to be further studied.] |
|
Comments 2: Secondly, while the paper establishes correlations between SOC components and functional groups, it lacks deeper mechanistic insights into the processes driving these changes. |
|
Response 2: Thank you for pointing this out. We agree with this comment. We added a discussion on the correlation between soil organic carbon components and organic carbon functional groups in lines 331-333. In addition, it is also stated in section 4.4 limitations that this study does lack mechanism research in the correlation part, so further research can be carried out. [There was a significant positive correlation between soil organic carbon content and microbial carbon content in the clay particles, indicating that the organic carbon in the clay particles mainly came from microorganisms.] Comments 3: Additionally, the MS could improve by discussing the long-term implications of vegetation restoration on carbon sequestration and ecosystem stability, which would enhance the overall relevance and impact of the study. Response 3: Thank you for pointing this out. We agree with this comment. Therefore, we have added a discussion of the long-term effects of vegetation restoration on carbon sequestration and ecosystem stability in lines 344-351. [Vegetation restoration diversifies plant community structure through the princi-ples of niche differentiation and complementarization, and increases litter quantity and input types to cause changes in the soil carbon pool (Gurmessa, et al. 2021). After veg-etation restoration, the increased input of root exudes, such as organic acids, the main component of root exudes, can not only stimulate microbial activity to promote the turnover of soil organic carbon by inducing an excitation effect but also change the structure of soil organic carbon by affecting the complexation and dissolution of organic carbon (Keiluweit, et al. 2015).] |
|
Comments 4: i would also suggest the authors can raad cite some good papers in the revised version if possible or applicable. DOI: 10.3390/rs15235571; DOI: 10.3390/rs13071279; DOI: 10.1016/j.agee.2022.108180; DOI: 10.1109/JSTARS.2023.3322640ï¼› Response 4: Thank you very much for your suggestions. I have read these four excellent papers in detail, and their ideas and innovations are very novel. However, we think that they are not relevant to our research, so we have not made any changes for the time being. Comments 5: furthermore, the ms is heavily focused on the scientific analysis of SOC components, but it could benefit from a more explicit discussion on the practical implications of the findings. For instance, how can these results inform land management practices, carbon sequestration strategies, or policy decisions related to vegetation restoration in the Loess Plateau or similar regions? in concluison, pl note that the authors can highlight the novel contributions of this study, particularly in terms of the methodological approach (e.g., use of FTIR) or the specific findings related to carbon sequestration mechanisms in the Loess Plateau. Response 5: Thank you for pointing this out. We agree with this comment. We add the practical significance of the results of this study in lines 95-97. In addition, new contributions of this study in methods are added in lines 408-409 of the conclusion. [In order to provide a theoretical reference for the change mechanism of aggregate or-ganic carbon in the fragile habitat area of the Loess Plateau, and provide a scientific basis for the sustainable management of agroforestry system in the Loess Plateau.] [In addition, FTIR technology has made a new contribution to carbon sequestration in the Loess Plateau.] 2. Response to Comments on the Quality of English Language |
|
Point 1: line 11 change into the turnover and properties of carbon within various aggregates change |
|
Response 1: However, during vegetation restoration, the turnover and properties of carbon within various aggregates changed. Point 2: line 12-13 consider simplifying for readability. Response 2: The effects of plant source carbon input on surface soil and subsurface soil may be different. Thus, the characteristics of carbon components in aggregates are affected. Point 3: line 35-36 change into "Soil aggregate" to "Soil aggregates are the basic units and components of soil organic carbon, closely related to soil organic carbon content.” Response 3: Soil aggregates are the basic units and components of soil organic carbon, closely related to soil organic carbon content. Point 4: Line 52-54 simplify the sentence Response 4: Therefore, the aim of this study was to investigate the effect of plant source carbon on the change and turnover of organic carbon components in aggregates. It provides sci-entific basis for the fixation mechanism of organic carbon in soil aggregates in fragile habitats of the Loess Plateau. Point 5: Line 304 consider changing "big" to "large" for consistency Response 5: The stability of large aggregates rose with vegetation recovery in the three soil layers, while the stability of clay particles decreased as vegetation recovery time in-creased (S1). |

Reviewer 2 Report
Comments and Suggestions for Authors
The manuscript is well-written and adequately supported by figures and tables.
After reviewing the entire text, I have a few minor suggestions and comments:
-
Is the study area 3x2 km?
-
In Figure 1, what do a, b, and c represent? Additionally, are the 0-20 cm, 20-40 cm, etc., the depths/thicknesses of the soil layers from the surface? Could you provide an explanation for the high content of RP32 in each panel? Also, the definition of "locust" years (16, 22, 32, and 47) seems to be missing from the data section, as the term "locust" is not mentioned.
-
The conclusion section feels somewhat shallow, as it lacks specific data on the increase or decrease in the content of the aromatic functional group. It might be valuable to include specific figures for these changes.
Author Response
|
1. Point-by-point response to Comments and Suggestions for Authors |
|
Comments 1: Is the study area 3x2 km? |
|
Response 1: Thank you for your question. In this study, three standard sampling areas of 20 m×20 m were randomly set in each sample plot. |
|
Comments 2: In Figure 1, what do a, b, and c represent? |
|
Response 2: Thank you for your question. Lowercase letters a,b,c indicate significant differences between plots of different afforestation years (P<0.05) . Comments 3: Additionally, are the 0-20 cm, 20-40 cm, etc., the depths/thicknesses of the soil layers from the surface? Response 3: Thank you for your question. The 0-20cm soil layer is 20cm away from the surface, the 20-40cm soil layer is 20-40cm away from the surface, and the 40-60cm soil layer is 40-60cm away from the surface. |
|
Comments 4: Could you provide an explanation for the high content of RP32 in each panel? Response 4: Thank you for your question. After 32 years of farmland return, the vegetation recovery was in the middle stage, and the litter input reached the peak, the microbial activity was the most active, so the organic carbon content was high. Comments 5: Also, the definition of "locust" years (16, 22, 32, and 47) seems to be missing from the data section, as the term "locust" is not mentioned. Response 5: Thank you for your question. Thank you for pointing out our mistake. RP we are meant to represent the Robinia pseudoacacia forest, and RP16 is the Robinia pseudoacacia forest for 16 years, RP32 is the Robinia pseudoacacia forest for 32 years, and so on. We have made corresponding changes to the comments under Figure 1 and Figure 2, please have a look. [Note: FL, RP16, RP22, RP32, and RP47 in the figure represent farmland, Robinia pseudoacacia forest 16 years, Robinia pseudoacacia forest 22 years, Robinia pseudoacacia forest 32 years, and Robinia pseudoacacia forest 47 years, respectively.] Comments 6: The conclusion section feels somewhat shallow, as it lacks specific data on the increase or decrease in the content of the aromatic functional group. It might be valuable to include specific figures for these changes. Response 6: Thank you for your advice. Due to vegetation restoration, the relative content of aromatic functional groups increased in soil aggregate carbon of every particle size, which was not as clear as the content of organic carbon. Therefore, it is not easy to repeat the specific increase, so we think it is a little tedious in the conclusion.
|
